# Branched flows of flexural elastic waves in non-uniform cylindrical shells

**Kevin Jose[1], Neil Ferguson[1], Atul Bhaskar[2]***

**1** Faculty of Engineering and Physical Sciences, University of Southampton, Southampton, United Kingdom,
**2** Department of Mechanical Engineering, University of Sheffield, Sheffield, United Kingdom

\* a.bhaskar@sheffield.ac.uk

**Data Availability Statement:** The data has been uploaded to the public GitHub repository: https://github.com/kevinjose/branchedFlowShell.

**Funding:** This work was supported by EU's Horizon 2020 programme under the Marie

## Abstract

Propagation of elastic waves along the axis of cylindrical shells is of great current interest due to their ubiquitous presence and technological importance. Geometric imperfections and spatial variations of properties are inevitable in such structures. Here we report the existence of branched flows of flexural waves in such waveguides. The location of high amplitude motion, away from the launch location, scales as a power law with respect to the variance, and linearly with respect to the correlation length of the spatial variation in the bending stiffness. These scaling laws are then theoretically derived from the ray equations. Numerical integration of the ray equations also exhibit this behaviour—consistent with finite element numerical simulations as well as the theoretically derived scaling. There appears to be a universality for the exponents in the scaling with respect to similar observations in the past for waves in other physical contexts, as well as dispersive flexural waves in elastic plates.

## Introduction

Waves propagating through heterogeneous media with spatially correlated randomness show a peculiar behaviour, known as branched flows [1], in a variety of physical contexts such as optics [2], microwaves [3], electron waves [4], tsunami waves [5], sound waves in ocean [6] etc. The phenomenon of branched flows of waves is characterised by the emergence of flow-like patterns with spatial branching. Emergence of branching is also associated with the occurrence of focusing events or caustics which leads to regions of high amplitude. Additionally, the expected distance $\langle l_f \rangle$—of the first of such focusing events from the point of launch—scales as

$$\langle l_f \rangle \propto L_c \langle h^2 \rangle^{-1/3}, \tag{1}$$

where $\langle \cdot \rangle$ signifies the mean, $L_c$ is the correlation length of the isotropic randomness field and $\langle h^2 \rangle$ is a non-dimensional measure of the severity of the randomness; $h$ is defined more rigorously below.

In the context of elastic waves, we recently showed the existence of branched flows and the associated scaling law in thin elastic plates [7]. This raises a natural question about the

Sklodowska-Curie scheme for doctoral training (Agreement No. 765636, Website: https://marie-sklodowska-curie-actions.ec.europa.eu/). Through this grant, A.B. & N.F. received research funding, and K.J. received a doctoral fellowship. The funders had no role in study design, data collection and analysis, decision to publish, or preparation of the manuscript.

**Competing interests:** The authors have declared that no competing interests exist.

universality of branched flows in mechanical waves carried by elastic structures. Pipes and elastic tubes are ubiquitous due to the ease of their fabrication and their frequent use to transport fluids through them. Many musical instruments, especially wind instruments, make use of cylindrical shells as acoustic wave guides. Cylindrical shells also are found in many practical applications such as health monitoring of buried gas pipes, where they act as waveguides for elastic waves. Cylindrical geometry affords a practical context of an elastic waveguide to study branched flows without reflections from the edges, unlike a wave bearing plate strip of finite width [7] where reflections from the two edges parallel to the main direction of propagation are inevitable. Further, the need for contrived periodic boundary conditions, as previously implemented in studies concerning propagation through random media in other branches of physics [8], is obviated. An cylindrical elastic surface wraps circumferentially onto itself, so it does not have any reflecting boundary running parallel to it axis, the main propagation direction; thus it naturally provides propagation space that eliminates reflections azimuthally.

Waves in elastic shells have attracted the attention of dynamicists for sometime [9, 10]. Pioneering work on the statics of cylindrical shells was carried out by Donnell [11]. This has been extended, for example, by Yu [12, 13] and Naghdi & Cooper [14], to derive the complete equations of motion. Numerous formulations of the dynamics of shells, of varying degrees of accuracy —often differing in the kinematic assumptions, exist; these have been summarized by Greenspon [10] and Leissa [15]. Propagating waves [16–18], normal modes [12, 13], shells under random excitation [19] have been studied theoretically, computationally [20] and experimentally [21]. The propagation behaviour of plane waves in the presence of inevitable manufacturing tolerances has not been studied in any detail so far. Here we examine the the effect of such spatial non-homogeneities and explore the emergence of channels of energy flow in such elastic waveguides.

Asymptotic approaches in elastic shells have been used frequently. A formal treatment of shell dynamics, after considering two parameters representing a length scale and a time scale, can be found in the works of Kaplunov et al. (see, e.g., [22]). Waves that propagate in one direction but are localised in another, i.e. exponentially decaying (e.g. Rayleigh waves [23] in semi-infinite medium) have been reported in the past (for example, the so called Konenkov [24] waves travelling at the edge of a flat plate, or other related waves in shells, see, e.g., Mikhasev [25]). This class of localised waves are also known as trapped waves, and often appear as a consequence of the boundary conditions (e.g. free-edge of an elastic plate). By contrast, the localised propagation via "channels" of propagation in a random medium, and reported here, are a consequence of the heterogeneity of the medium.

We consider the propagation of flexural waves through a hollow cylindrical waveguide when the wavelength of interest is much shorter than the correlation length of the heterogeneity of properties, e.g., thickness or material stiffness. The tubular cross-section undergoes breathing displacement that is radial, as a deformation pattern propagates axially as a wave. The nominal thickness of the elastic cylinder is much smaller than the wavelength of interest, in order to justify ignoring shear through the thickness in our analyses. Consider a cylinder of non-uniform thickness with the axis along the $x$-direction; the circumferential direction along $s$ (with units of length) and thickness $H(x, s)$ about the nominal cylindrical surface. The thickness of the hollow elastic cylinder has the form $H(x, s) = H_0(1 − h(x, s))$ where $h(x, s)$ is a smoothly varying random field with an isotropic auto-correlation function; the correlation length is $L_c$, and $\langle h \rangle = 0$, $\langle h^2 \rangle \ll 1$ (see Fig 1).

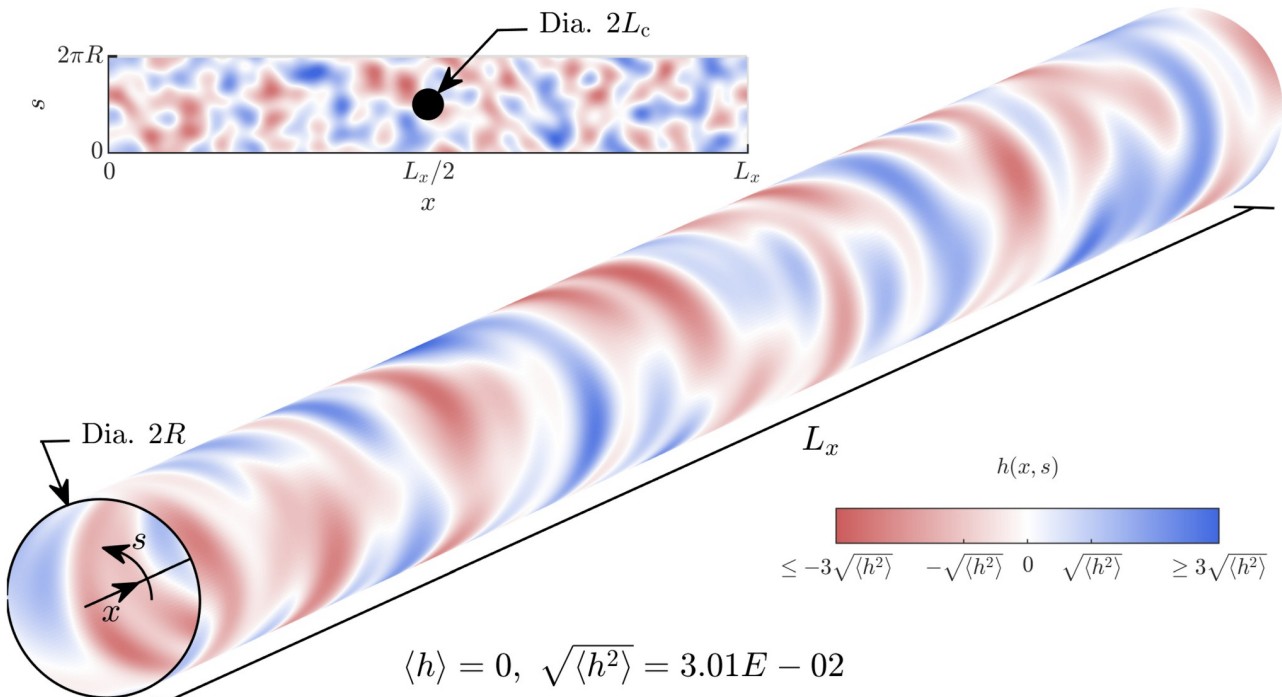

**Fig 1. Spatial variation of thickness over the surface of a cylindrical waveguide.** The radius of the cylindrical shell is $R$, axial length $L_x$, thickness $H$ that varies spatially as per $H(x, s) = H_0(1 − h(x, s))$, where $\langle h \rangle = 0$, $\langle h^2 \rangle \ll 1$. Heatmap shows the spatial variation of $h(x, s)$. Inset on the top left is an "unwrapped" view of $h(x, s)$. In the middle of the inset, a circle of radius $L_c$, equal to the correlation length, is shown.

## Results

Consider an initially plane wavefront of predominantly one wavelength propagating axially through a thin cylindrical shell. The assumption of slow spatial variation of properties enables the simplification of the wave elastodynamics to a set of ray equations using the eikonal/ WKB approximations [26, 27]. They are further simplified using the paraxial approximation [28], permitted by the weak scattering nature of the problem, which asserts a predominantly axial direction of the wave vector. These ray equations are then used to analytically derive the scaling law (Eq 1) relating the position of focusing and the severity of the non-homogeneity. We also numerically integrate the ray equations to investigate the emergence of branched flow and to validate the theoretically derived scaling law. The same scaling is also probed using finite element simulations which capture the complete wave elastodynamics, including dispersion.

### Ray equations

**Thin shell theory.** Plane waves of predominantly single wavelength $k_0$ are launched at the left end of the elastic waveguide. Rays are fully described by four quantities: spatial variables $x$, $s$ indicating the location of the ray along the axis and circumference respectively, and $k_x$, $k_s$, the wavenumber components in the axial and circumferential directions respectively. The spatial variables are non-dimensionalised with respect to the nominal radius, i.e. $\tilde{x} = x/R, \tilde{s} = s/R$, whereas the wavenumbers are non-dimensionalised with respect to the initial wavenumber $\tilde{k}_x = k_x/k_0, \ \tilde{k}_s = k_s/k_0$. Given the parameter regime of interest, (thin shell, small curvature, and slowly varying parameters) we can use the dispersion relation given by Pierce [16] to

obtain the non-dimensionalized ray equations

$$\partial_\tau \tilde{x} = \tilde{k}_x \left( \tilde{k}_x^2 + \tilde{k}_s^2 - \alpha \frac{\tilde{k}_x^2 \tilde{k}_s^2}{\left(\tilde{k}_s^2 + \tilde{k}_x^2\right)^3} \right),$$

$$\partial_\tau \tilde{s} = \tilde{k}_s \left( \tilde{k}_x^2 + \tilde{k}_s^2 + \alpha \frac{\tilde{k}_x^4}{\left(\tilde{k}_s^2 + \tilde{k}_x^2\right)^3} \right), \tag{2}$$

$$\begin{bmatrix} \partial_\tau \tilde{k}_x \\ \partial_\tau \tilde{k}_s \end{bmatrix} = \frac{1}{2} (\tilde{k}_x^2 + \tilde{k}_s^2)^2 \tilde{\nabla} \tilde{h},$$

where $\alpha = 12(v^2 - 1)\frac{\gamma^2}{\epsilon^2}$, $\gamma = (k_0 R)^{-1}$, $\epsilon = H_0 k_0$. Additionally, $\tilde{h}(x/R, s/R) = h(x, s)$ and $\tilde{\nabla} = [\partial_{\tilde{x}} \ \partial_{\tilde{s}}]^T$. Here, $\tau$ is an arbitrary time variable, consistent with ray approximations. The initial condition is $\tilde{x} = 0$, $\tilde{k}_x = 1$, $\tilde{k}_s = 0$ and $\tilde{k}_s \in [0, 2\pi R)$ depending on the circumferential location of the ray's starting point.

**Paraxial ray equations.** In the regime of weak scattering studied here, the thin shell ray equations can be simplified. $\tilde{k}_x$ and $\tilde{k}_s$, the wavenumbers in the axial and circumferential directions respectively, are not expected to vary substantially from their initial values of 1 and 0 respectively, as the initially launched plane wave is purely axial, and it remains predominantly axial. For weak scattering of initially plane waves, it is customary to make the simplifying assumption that the wave-number does not change at all in the main propagation direction. This can be achieved by setting $\partial_\tau \tilde{k}_x = 0$, $\tilde{k}_x = 1$ in Eq 2, which is the essence of the paraxial approximation. Finally, dropping all $\tilde{k}_s$ terms compared to $\mathcal{O}(1)$ terms, the ray equations become

$$\partial_\tau \tilde{x} = 1, \quad \partial_\tau \tilde{s} = \tilde{k}_s(1 + \alpha), \quad \partial_\tau \tilde{k}_x = 0, \quad \partial_\tau \tilde{k}_s = \frac{1}{2} \partial_{\tilde{s}} \tilde{h}. \tag{3}$$

The approximations made to obtain Eq 3 from the full thin shell ray equation use arguments about the physics of the problem. Nonetheless, the validity of these assumptions is further confirmed later here by numerical integration of ray equations.

Fig 2 (left) shows a comparison between the rays obtained from numerical integration of the full thin shell ray equations (Eq 2), and the simplified equations obtained following the paraxial approximation (Eq 3) for increasing values of $\langle h^2 \rangle$. The rays have been plotted "unwrapped" in the left column of the figure, where 300 rays equi-spaced along the circumference at the left end are launched as they curve and veer downstream due to scattering. Transmission behavior on the cylindrical surface, is shown in Fig 2 (middle) where interesting spiral structures with no preferred handedness, as expected, are observed. Such structures are not present in the case of branched flow of flexural waves in thin plates [7] due to the presence of reflecting boundaries. The rays as computed using thin shells theory versus that using the paraxial approximation look fairly similar, confirming the validity of the paraxial approximation. At the time instant for which the simulation is terminated, all rays have reached the right edge in case of the paraxial approximation, unlike the rays obtained from integrating the thin shell ray equations. This is a consequence of assuming $\partial_\tau \tilde{k}_x = 0$ under the paraxial assumption. Most importantly, it can be seen that the spatial location of the first caustic, indicated by circular and cross-shaped markers, is approximately the same from both sets of ray equations, except for the highest levels of $\langle h^2 \rangle$ shown here.

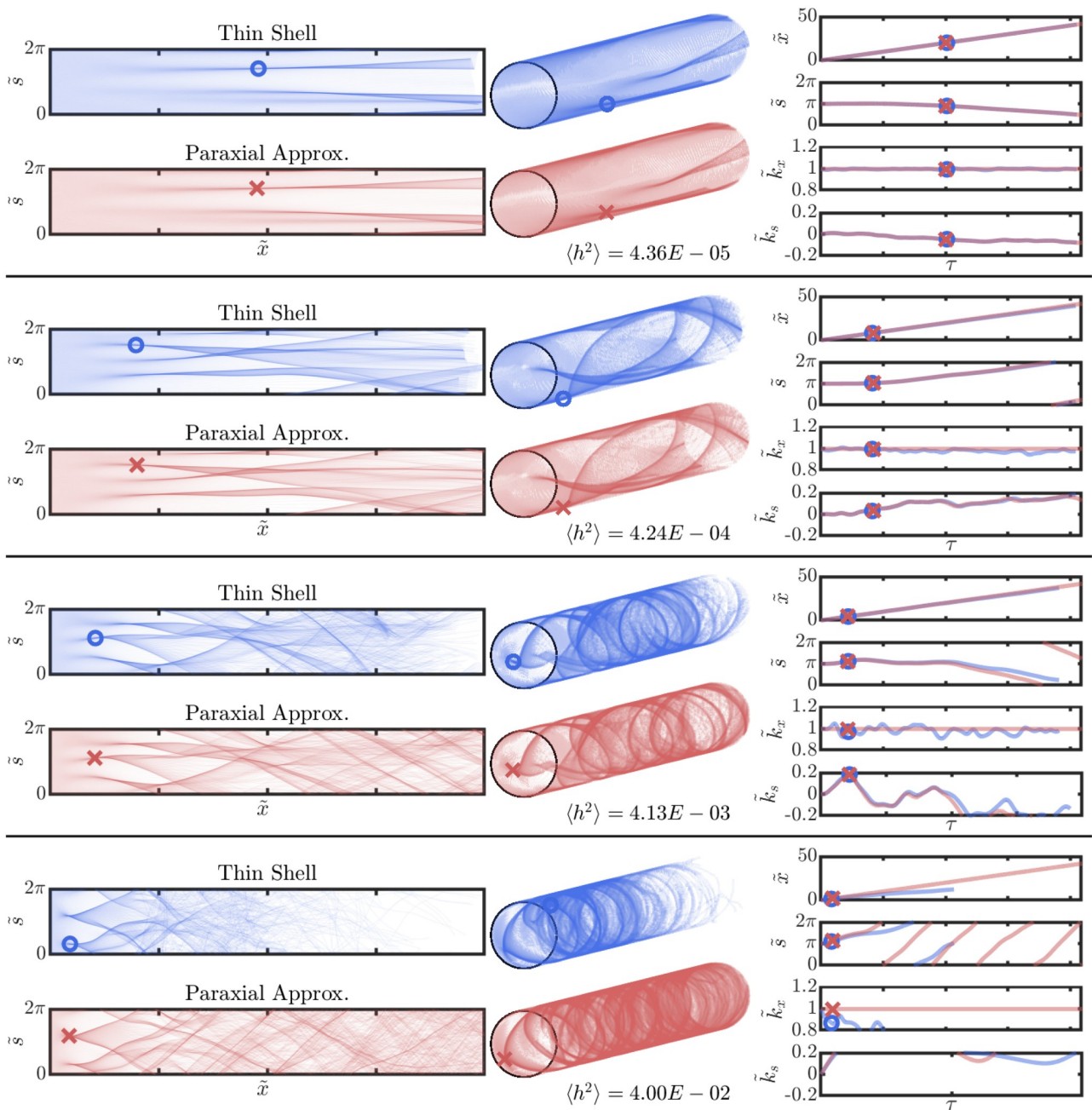

**Fig 2. Comparison of thin shell ray equations and equations obtained from making the paraxial approximation.** Left: Ray propagation using full thin shell theory and paraxial approximation of thin shell theory. Circular and cross markers indicate the location of the first caustic. The $\tilde{s}$ axis has been "unwrapped" for representational purposes. Middle: Same ray propagation shown on the cylindrical shell. Right: Plot of the temporal evolution of quantities describing the one of the rays. Here, markers indicate *temporal* location of the first caustic. It can be seen that, at higher values of $\langle h^2 \rangle$, the location of the first caustic detected from the paraxial approximation differs from thin shell.

In Fig 2 (right), the time evolution of the four quantities describing one ray is plotted. The values of $\tilde{x}$, $\tilde{s}$, $\tilde{k}_s$ obtained from the two ray formulations are in excellent agreement. $\tilde{k}_x$ has a constant value in the paraxial approximation as stated earlier. While it does not show the variation with time that the thin shell case shows, note that the values do not change appreciably

from 1. Regardless, the first caustic is detected by looking for the instant where the $\tilde{s} - \tilde{k}_s$ curve becomes locally two valued (details below) and is not dependant on $\tilde{k}_x$ directly. In Fig 2 (right), markers indicate temporal location of the first caustic.

The paraxial approximation becomes more inaccurate as time passes. However, since we are interested only in the location of the first caustic, and they tend to appear fairly early, this eventual drift is inconsequential. The paraxial approximation also breaks down faster when the value of $\langle h^2 \rangle$ is higher, i.e. when the weak scattering assumption starts breaking down. However, this is partially counteracted by the fact that when the value of $\langle h^2 \rangle$ is higher, the first caustic appear earlier. Nonetheless, the progressive degradation of the paraxial approximation with increase in $\langle h^2 \rangle$ can be seen in Fig 2.

We will now use these ray equations to derive the scaling of $\langle l_f \rangle$ analytically. We will also numerically integrate them to study the scaling. The ray equations Eq 2 (and, therefore, Eq 3 too) are obtained from simplified dispersion relation for flexural waves in a thin shell. At the higher $\langle h^2 \rangle$, some rays are expected to completely back-scatter due to the severity of randomness. This is not captured by Eq 2. In the S1 File, we derive the ray equations starting from the governing equations of the displacements of a thin cylindrical shell. These ray equations are more sophisticated and show the expected back scattering at higher values of $\langle h^2 \rangle$. However, after applying the paraxial approximation, the resultant set of ray equations show very similar results to Eq 3. The analytical and numerical validation of the scaling law Eq 1 using these ray equations is also shown in the S1 File.

## Scaling law: From analysis, ray equations, and finite element elastodynamics

We use the ray equations obtained after making the paraxial approximations Eq 3, and following a process very similar to other branched flow works [5, 7] we obtain, $\langle l_f \rangle \sim L_c \langle h^2 \rangle^{-1/3}$ barring an extremely weak dependence of the proportionality constant on $\gamma$ and $\epsilon$ (S1 File).

The scaling of the location of the first caustic resulting from the analysis of ray equations can also be obtained numerically from: (i) numerical integration of the ray equations, and (ii) finite element elastodynamic simulations. The emergence of branched flows is clearly visible from finite element elastodynamics simulations; an example of which is shown in Fig 3. 'Snapshots' of the temporal evolution are shown in (a → b → c → d) of an initially plane wave front, as it propagates along the elastic cylinder with non-uniform thickness. The entire domain is shown on the top of each panel and regions of high amplitude, indicated by colored lines, are zoomed into and shown on the bottom of each panel. The radial displacement has been greatly exaggerated for representational purposes. Note that the absolute value of displacement can be scaled arbitrarily since we are considering the linear regime. The initially plane wave front (a) splits into distinct branches (b) leading to regions of extreme amplitudes. As the wavefront propagates further (c, d) more branching is observed. The widening of the wavefront as expected because of the dispersive nature of flexural waves in shells as seen here.

Using the approach of detecting the first caustic described in the Methods section, the scaling of location of the first caustic with the statistical properties of the random field and the geometry of the cylinder are now explored. Fig 4 shows results for the distance of the first focusing event from the point of launch as a function of the severity of randomness for 800 realisations of the shells of correlated randomness. The mean for each value of $\langle h^2 \rangle$ is shown by a blue square marker. The figure of the left is from the numerical integration of ray equations whereas that on the right is from FE simulations, both showing good agreement with the scaling $\langle l_f \rangle \sim \langle h^2 \rangle^{-1/3}$.

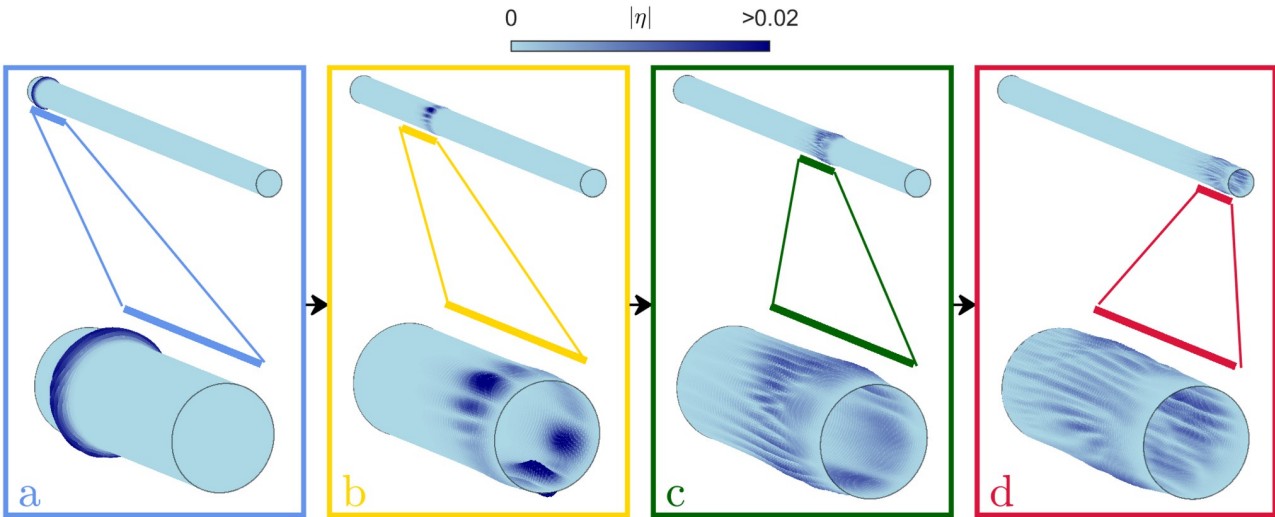

**Fig 3. Emergence of branched flow in for an initially plane wavefront in a thin cylinder of non-uniform thickness.** Temporal evolution (a → b → c → d) of an initially plane flexural wave front in a thin elastic cylinder of non-uniform thickness. The full cylinder has been shown on the top of each panel. The regions of high amplitude at each time instant is indicated with colored lines and they been zoomed into and shown at the bottom of each panel. The emergence of branching leading to locations of extreme amplitudes and widening of the wavefront consistent with the dispersive character of this class of elastic waves is also observed.

The linear scaling of the location of focusing with the correlation length $\langle l_f \rangle \propto L_c$ is also validated using the two numerical methods, see Fig 5. Using the curve obtained for $L_c/L_c^{\mathrm{ref}} = 1$ as reference, the prediction assuming the linear scaling with $L_c$ is shown using dashed lines. It is clear that the actual simulations (markers) agree with this prediction both for ray simulations as well as finite elements elastodynamics. In the results from FE elastodynamics simulations, the scaling seems to diverge from the prediction at higher values of $\langle h^2 \rangle$; this is expected since the weak scattering assumption breaks down at these values of $\langle h^2 \rangle$. This does not happen in the simulations using numerical integration of ray equations since we use the formulation obtained from applying the paraxial approximation which assumes weak scattering. The linear scaling can also be inferred from dimensional arguments that the only length scale in the problem is the correlation length.

The insensitivity of $\langle l_f \rangle$ to wavelength (for $\lambda \ll L_c$) is confirmed by the two numerical approaches, see Fig 6. The expected breakdown of the scaling due to strong scattering is seen here for the FE elastodynamics simulations. Finally, using ray simulations, we verified that $\langle l_f \rangle$

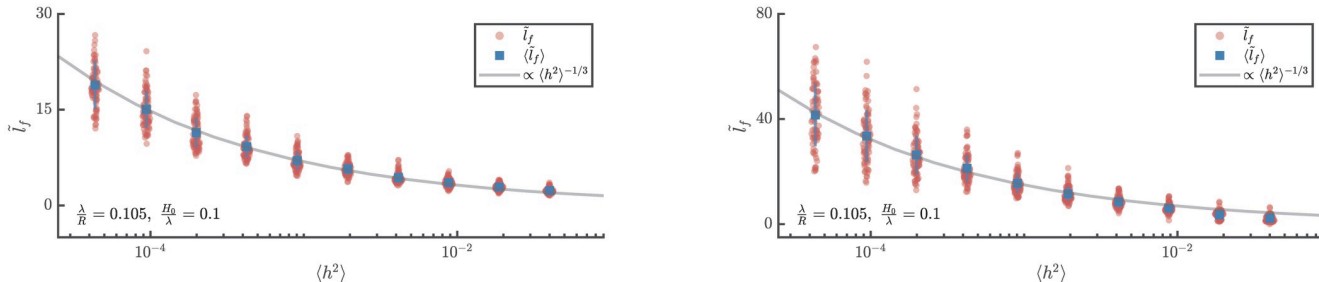

**Fig 4. Scaling of location of the expected location of the first focusing event with "severity" of randomness.** Locations of first caustic as obtained from numerical ray integration (left) and FE elastodynamics simulations (right). Circular markers indicate locations of first caustic from individual simulations. Square markers show the expected location of the first caustic. Vertical lines indicate 2 standard deviation (centered around the mean). The expected locations of the first caustic show the expected power law scaling with $\langle h^2 \rangle$.

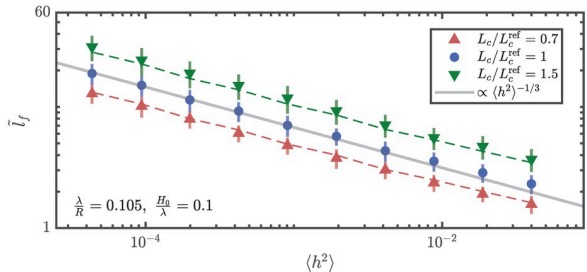
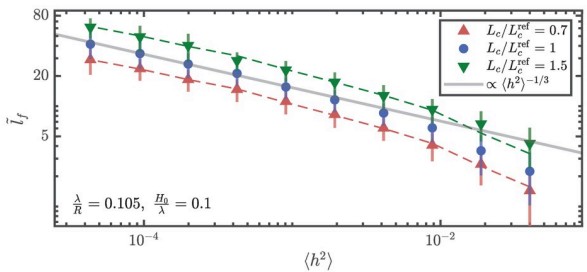

**Fig 5. Scaling of the expected location of the first focusing event with correlation length of the randomness.** Expected locations of the first caustic from numerical ray integration (left) and FE elastodynamics simulations (right) for different correlation lengths. Taking the curve corresponding to $L_c/L_c^{\mathrm{ref}} = 1$ to be the reference, dashed lines show the predicted behavior at other correlation lengths assuming $\langle l_f \rangle \propto L_c^1$. The actual simulations agree well this prediction hence confirming the linear scaling.

is insensitive to variations of the radius of the cylinder in the shallow cylinder regime $\lambda \ll R$, see Fig 7. The same could not be carried out using FE simulations due to computational constraints. Note that, since $\tilde{l}_f$ is non-dimensionalised with respect to the radius, a nominal radius $R_0$ is used to scale $\tilde{l}_f$ appropriately when studying (in Fig 7) the effect of changing radius.

## Discussions

The present study extends our previous observations for non-homogeneous elastic plates. The presence of branched flows in thin elastic shells suggests the robustness of the phenomenon across physics and also geometry of the wave-bearing media. Further, the scaling of the location of the first focusing event from the point of launch of the waves appears to have universality in terms of the exponent on the measure of heterogeneity.

The ray dynamics approximation of this wave propagation problem presents a numerically inexpensive way to explore the essential physics of this phenomenon without an associated high computational cost. Nonetheless, some of the characteristics of this wave-bearing media are not captured by the ray dynamics approximation. This is why we complemented it with full wave elastodynamics simulations using FEM. While, FEM is of much higher fidelity than numerical ray dynamics, some high frequency/short wavelength regime behaviour may still have been missed out due to the limitation on how fine the meshing could realistically be.

The elastic medium as well as the randomness considered here are isotropic. When the randomness is anisotropic, the directionality of randomness is likely to affect the branched flows,

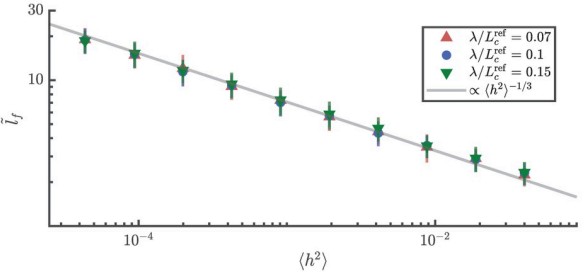
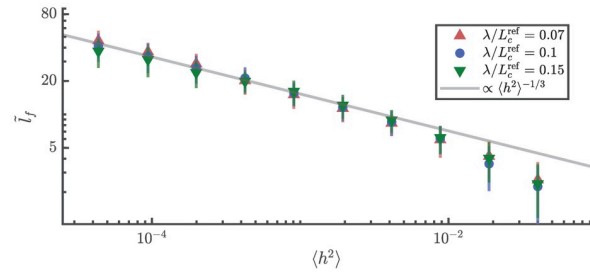

**Fig 6. Scaling of the expected location of the first focusing event wavelength of the initial wavefront.** Expected locations of the first caustic from numerical ray integration (left) and FE elastodynamics simulations (right) for different wavelengths. As long as $\lambda \ll L_c$, $\langle \tilde{l}_f \rangle$ is independent of wavelength. The power law scaling seems to break down at higher $\langle h^2 \rangle$, especially in FE simulations. This is consistent with the expectation that the scaling holds only for weak scattering and higher $\langle h^2 \rangle$ corresponds to higher scattering.

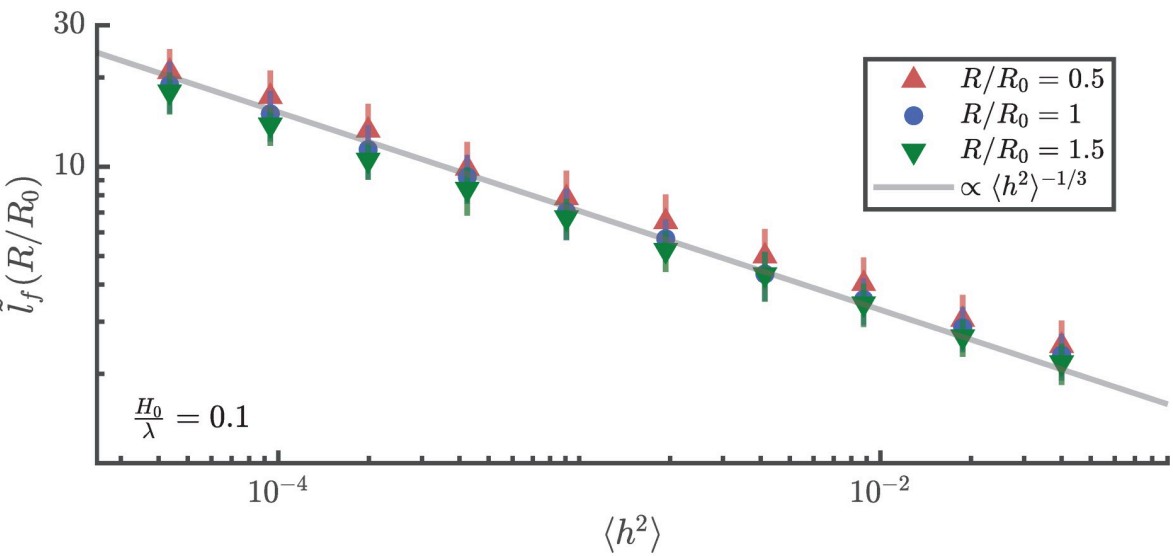

**Fig 7. Scaling of the expected location of the first focusing event with the radius of the cylinder.** Scaling of the expected locations of the first caustic with radii from numerical ray integration. The expected location of the first caustic is independent of the radius in the parameter ranges of interest.

and intuitively a directional bias is expected. Likewise, directionality in the medium itself would change the governing equation of motion, via the constitutive relationship and the wave-speed on the surface of a cylindrical shell will have a directional dependence. This could be of practical interest to many engineering structures where orthotropic shells or laminates with orthotropic layers are commonly used. We would expect a chirality in the branched flows under such situations. However, we are not concerned with these aspects in the present work.

The findings of the paper could have a more general implication, but at this stage such suggestions are at best speculative. For example, one could imagine the emergence of branched flows of waves in infinite elastic plates radially spreading from a source driven at a point. Branched flow for similar geometry in water waves [29] have been reported. The presence of the phenomenon reported for isotropic shells is expected to be robust to other isotropic materials, because regardless of the material in question, the properties are described by just two elastic constants. Material properties do not appear in any of the scaling concerning branched flows, instead the non-dimensional level of randomness does. Branched flows in anisotropic elastic media remains an open question.

The findings of the present work may have interesting implications to acoustic condition monitoring of elastic pipes such as those used by Pipeline Inspection Gadgets (or Guages, PIGs for short; the process being known as "pigging"), where inhomogeneity in the elastic shell to be monitored as well as its geological surroundings is inevitable. Insights into the relationship between locations of caustics and the "severity" of randomness in the elastic properties can also potentially be applied to the metrology of thin cylindrical shells.

## Conclusions

There appears to be a universality of branched flows in the propagation of elastic waves through random media. Following our previously reported results on branched flow in elastic plates [7], here we demonstrate analogous behaviour for shells with correlated random properties. The ray equations for flexural waves in a thin elastic cylinder are derived. A paraxial approximation, permitted by the parameter regimes of interest, is applied to the ray equations.

This is used to analytically demonstrate that the expected location of the first caustic in shallow cylinders shows the scaling $\langle l_f \rangle \propto L_c \langle h^2 \rangle^{-1/3}$. This scaling was then corroborated using numerical integration of ray equations and full FE elastodynamic simulations.

An immediate extension of this work on cylindrical elastic shells would be to explore the dependence of the scaling of the first caustic with the radius for shells with *appreciable* curvature (i.e small radius). We are unable to do so in this work since the requirement of $2\pi R \gtrsim L_c$ limits how small the radius can be. This can be remedied by using anisotropic randomness which would enable one to reduce the radius by using a smaller correlation length in the circumferential direction. There may be an elegant scaling of $\langle l_f \rangle$ with radius in this parameter regime. The existing literature on branched flows in media with spatially anisotropic randomness [30] can be leveraged.

## Methods

### Detecting first focusing events

For numerical ray simulations, we use a method similar to that described in our earlier work on flat plate dynamics [7] to detect focusing events. In the ray picture, a caustic corresponds to the location where the density of rays becomes infinity. It can be shown that the "density of rays" at any given point will be inversely proportional to $\partial \tilde{s}/\partial \tilde{k}_{\tilde{s}}$. Therefore, caustics can be detected by finding locations where the $\tilde{k}_{\tilde{s}} - \tilde{s}$ curve folds over itself i.e. $\partial \tilde{s}/\partial \tilde{k}_{\tilde{s}} \to 0$. See [29, 31] for details. The first focusing event is, therefore, detected by finding the time and location when the $\tilde{k}_{\tilde{s}} - \tilde{s}$ curve folds over itself for the first time. This is done numerically by tracking the local slope of the curve and detecting locations when the slope become higher than $\pi/2$. The temporal evolution of the $\tilde{k}_{\tilde{s}} - \tilde{s}$ curve is plotted in Fig 8. The first time this curve folds over itself, signalling a caustic, is shown in red. Note that since, $\tilde{k}_{\tilde{s}}$ has values centered around 0, a constant positive value is added to it for representational purposes. This value is not used when numerically detecting the caustic.

In the wave picture, we do not detect the location of the caustic directly. We detect the caustic, instead, by locating high amplitude events, *a consequence* of caustics. From an implementation point of view, we use a very similar technique as the one used in existing branched flow literature [7] to detect the location of the first caustic from Finite Elements (FE) simulations. For each simulation, the displacement fields are used to construct the integrated intensity, $I(x, s) = \int_0^T \eta^2(x, s, t)\mathrm{d}t$. This is in turn used to construct the scintillation index, $S(x) = \langle I^2 \rangle_s / \langle I \rangle_s^2 - 1$. The location of the first significant peak of $S(x)$ indicates the location of the first caustic (see Fig 9).

### Remarks on numerical ray integration and FE elastodynamics

According to the paraxial approximation, rays travel at a constant speed in the main propagation direction. This wavespeed corresponds to the speed of propagation for the monochromatic wave being launched. This enables us to calculate the total time the rays would take to travel the entire length of the cylinder. The time stepping in the numerical ray simulations is set using this to ensure that there are 2000 time steps in the full length traversal. We can be certain that this time stepping is adequate since no variation is seen in the rays when the time steps are varied around 2000. Another way of being confident of the adequacy of the time stepping is to note that the time steps roughly corresponds the same number of steps the main propagation direction and 2000 steps is adequate to capture the features in the main propagation direction which are of the order of the correlation length i.e $(L_x/2000 \ll L_c)$. Note that we

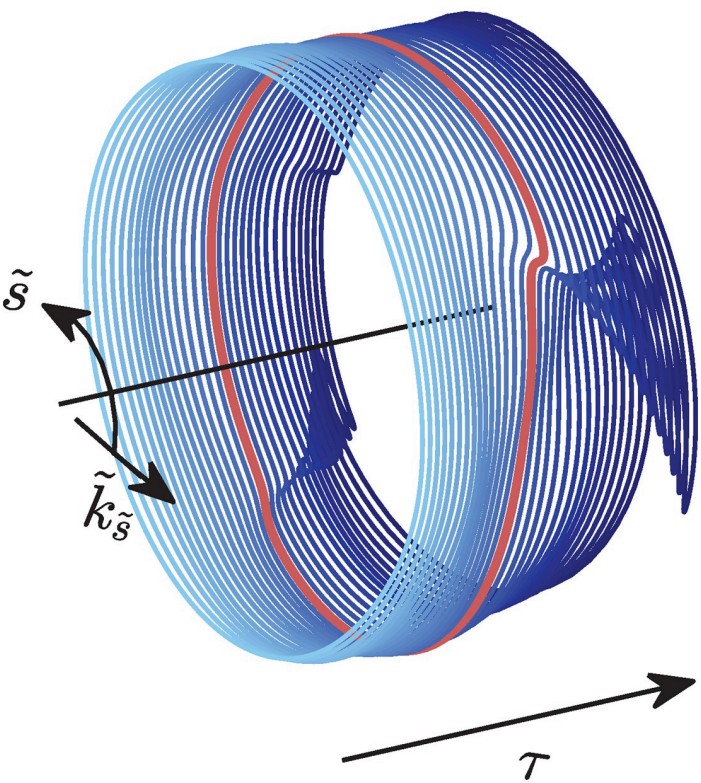

**Fig 8. Plot of the temporal evolution of the $\tilde{k}_{\tilde{s}} - \tilde{s}$ curve obtained from numerical ray simulations.** Since values of $\tilde{k}_{\tilde{s}}$ are centered around zero, a constant positive offset is added for the purpose of visualization. The first caustic is detected by finding the location where the $\tilde{k}_{\tilde{s}} - \tilde{s}$ curve folds over itself (shown in red).

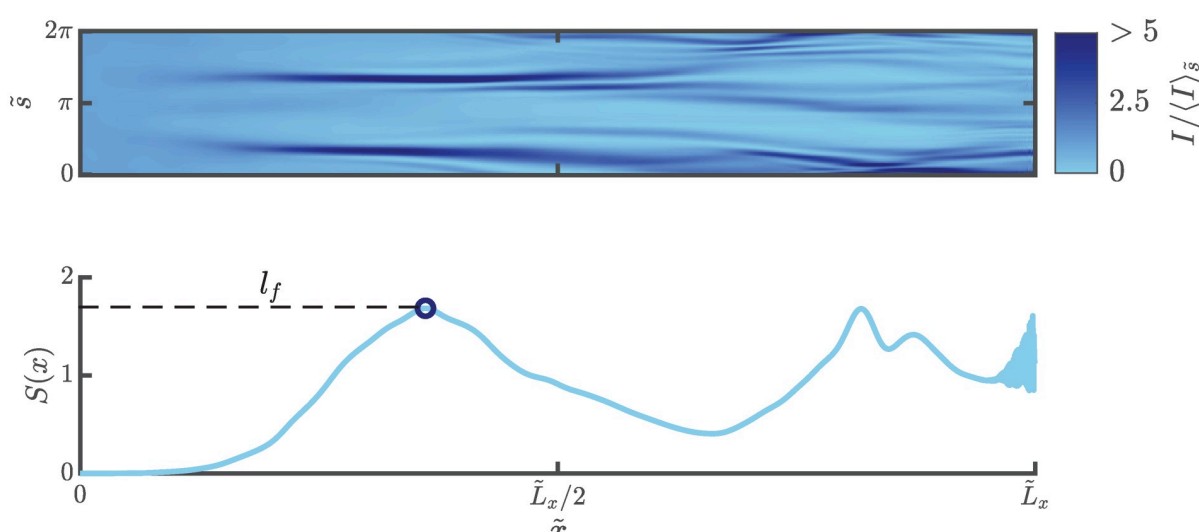

**Fig 9. Plot of integrated intensity (top) and scintillation index (bottom) of an exemplar FE simulation.** Note that the integrated intensity has been normalized along by the mean along the circumferential direction for visually emphasizing the locations of extreme amplitudes. The first prominent peak (circular marker) of the scintillation index ($SI(x)$) curve indicates the location of the first caustic ($l_f$).

are unable to use any other physical arguments to arrive at the time step since the time scaling for ray equations is arbitrary as shown earlier.

For the FE elastodynamics simulations, a combination of physical arguments and some trial and error is used to find the total time of the simulations. The time step is set to $\frac{\pi}{4\omega}$, this ensures that there are 8 points per cycle since the forcing has predominant frequency component of $\omega$. The cylinder is meshed using rectangular element which have the dimensions $\lambda/8$ and $3\lambda/8$ in the axial and circumferential directions respectively. Note that the meshing can be a bit coarser in the circumferential direction since the wave predominantly travels in the axial direction and therefore the variations along the circumference are modest and of length scales much longer than the wavelength. The element aspect ratio of 1:3 is typically inadvisable in general, however, since we are confident that the variations in the circumferential direction are modest and of longer length scales, this aspect ratio will not lead to ill conditioning or misestimation of results. The spatial discretization of $\lambda/8$ ensures that the spatial variation due to a wave of predominant wavelength $\lambda$ is adequately captured. Admittedly, since the system being studied is dispersive, wavelength shorter than $\lambda$ will also be excited and the chosen discretization may not model them well. The computational expense of these FE simulations must be emphasized here and this is the reason behind some compromises in the choices made during meshing.

It was ensured that the location of the detected caustics from numerical ray integration did not show any angular bias. Fig 10 show the angular distribution of caustics detected using numerical ray integration. No consistent and appreciable bias is visible. Angular bias can be introduced inadvertently from improper interpolation of the randomness field when conducting numerical ray integration.

Ray integration and FE simulation routines developed for our work on branched flows in elastic plates [7] were suitably modified for the elastic cylindrical domain under consideration. Most of the required modifications pertain to imposing the continuity condition as one went around the circumferential direction. The method detailed in [7] for generating random field

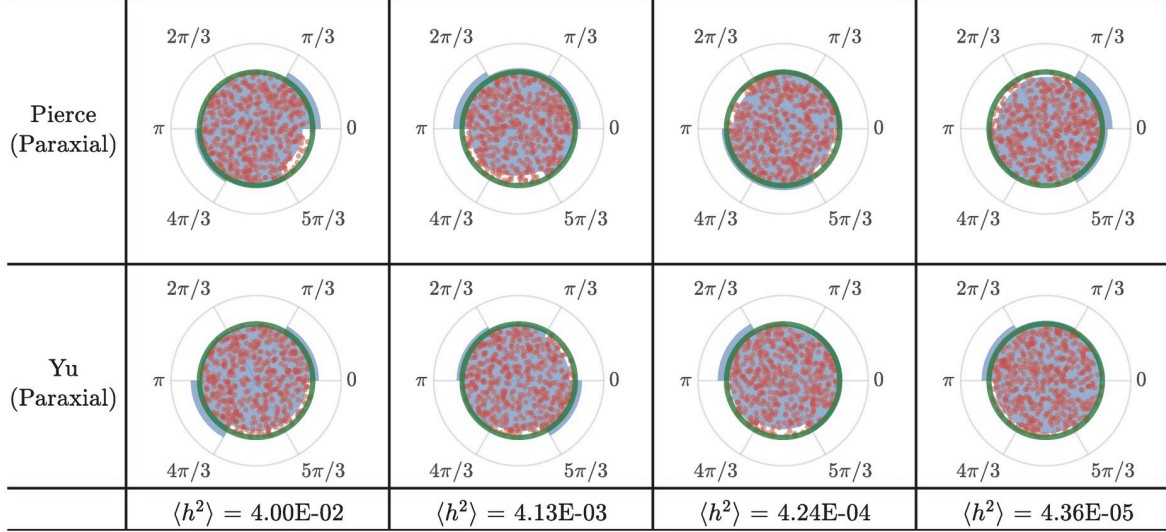

**Fig 10. Histograms of angular locations of the first focusing events obtained from numerical ray integration.** Polar histograms of the angular locations, shown in blue, show no consistent appreciable angular bias. The green circle indicates the outline of a polar histogram with perfectly zero angular bias. The individual data-points used to construct the histograms are shown with orange dots. Their angular location is the one obtained from simulations; their radial position in the above plots is randomized in the interest of visualization.

of specified correlation length, fortuitously, generates fields which are periodic along the parallel sides and hence, that code was ready to repurposed with minimal modification for generating $h(x, s)$ since the continuity requirement was automatically satisfied. The code snippet detecting caustics numerically during ray simulations was rewritten to respect continuity in the circumferential direction. The parameterized FE code was modified to generate the right circular cylindrical geometry. The code to export integrated intensities was also modified in light of the different coordinate system.

## Supporting information

**S1 File. Derivation of ray equations and scaling law from Yu's formulations of the equations of motion.**
(PDF)

**S1 Video. Animation showing emergence of branched flows of flexural waves in a cylindrical shell.** (Top) Animation showing the eventual random focusing and branched flow of an initially planar flexural wave front in an elastic cylinder. (Bottom) A zoomed in view of the wavefront. Emergence of branches and high amplitudes is clearly visible.
(AVI)

## Acknowledgments

Thanks are due to the University of Southampton for access to Iridis Compute Cluster; Claus Ibsen, Vestas aircoil for providing a practical context.

## Author Contributions

**Conceptualization:** Kevin Jose.

**Data curation:** Kevin Jose.

**Formal analysis:** Kevin Jose.

**Funding acquisition:** Neil Ferguson, Atul Bhaskar.

**Investigation:** Kevin Jose, Neil Ferguson, Atul Bhaskar.

**Methodology:** Kevin Jose, Neil Ferguson, Atul Bhaskar.

**Project administration:** Neil Ferguson, Atul Bhaskar.

**Resources:** Neil Ferguson, Atul Bhaskar.

**Software:** Kevin Jose.

**Supervision:** Neil Ferguson, Atul Bhaskar.

**Validation:** Kevin Jose.

**Visualization:** Kevin Jose.

**Writing – original draft:** Kevin Jose, Neil Ferguson, Atul Bhaskar.

**Writing – review & editing:** Kevin Jose, Neil Ferguson, Atul Bhaskar.

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
