## [Decision Letter · Decision Letter 0]

8 Mar 2023

PONE-D-23-04637Branched flows of flexural elastic waves in non-uniform cylindrical shellsPLOS ONE

Dear Dr. Bhaskar,

Thank you for submitting your manuscript to PLOS ONE. After careful consideration, we feel that it has merit but does not fully meet PLOS ONE’s publication criteria as it currently stands. Therefore, we invite you to submit a revised version of the manuscript that addresses the points raised during the review process.

I agree to the comments provided by the expert reviewers. Overall, the article meets the standard set  by Plos One and thus deserves to be published after addressing the observations raised by the reviewers.  

We look forward to receiving your revised manuscript.

Kind regards,

Rab Nawaz

Academic Editor

PLOS ONE

Journal Requirements:

"Thanks are due to the University of Southampton for access to Iridis Compute Cluster; 277

EU’s Horizon 2020 programme under Marie Sk lodowska-Curie scheme (Grant 278

Agreement No. 765636) for financial support; Claus Ibsen, Vestas aircoil for providing a 279

practical context."

"This work was supported by EU’s Horizon 2020 programme under the Marie Sklodowska-Curie scheme for doctoral training (Agreement No. 765636, Website: https://marie-sklodowska-curie-actions.ec.europa.eu/). Through this grant, A.B. & N.F. received research funding, and K.J. received a doctoral fellowship. The funders had no role in study design, data collection and analysis, decision to publish, or preparation of the manuscript."

Reviewers' comments:

Reviewer's Responses to Questions

**Comments to the Author**

1. Is the manuscript technically sound, and do the data support the conclusions?

Reviewer #1: Yes

Reviewer #2: Yes

2. Has the statistical analysis been performed appropriately and rigorously? 

Reviewer #1: Yes

Reviewer #2: Yes

3. Have the authors made all data underlying the findings in their manuscript fully available?

Reviewer #1: Yes

Reviewer #2: Yes

4. Is the manuscript presented in an intelligible fashion and written in standard English?

Reviewer #1: Yes

Reviewer #2: Yes

5. Review Comments to the Author

Reviewer #1: The manuscript is concerned with branched flows of flexural waves in non-uniform elastic cylindrical shells with slow spatial variation of properties. The scaling laws involving the expected distance between the point of launch and the first of the focusing events, the correlation length of the isotropic randomness field, and a non-dimensional measure of the degree of randomness, are derived theoretically from the ray equations. The ray equations are solved numerically to illustrate the emergence of branched flow and verify the theoretical prediction for the scaling law. Finite element simulations are also performed to present the full wave elastodynamics.

The manuscript is within the scope and of clear interest for the journal. This is a challenging but useful problem. I am happy to suggest its acceptance, after the authors address the comments listed below.

1. The state of art could be slightly better described. In particular, for delicate aspects of applying WKB method for thin shells see Mikhasev and Tovstik (Localized Dynamics of Thin-Walled Shells, 2020) and references therein; for classification of dynamic regimes in thin elastic shells and limitations of the theory see Kaplunov, Kossovich and Nolde (Dynamics of Thin Walled Elastic Bodies, 1998).

2. It would be good to explain about the first caustics in slightly more detail, possibly with more formulae?

3. Since the authors have previously studied branched flows of flexural elastic waves in plates [7], it would be good to compare what are the novel features emerging in shells?

4. In formulae (2) in brackets what is the purpose of putting numerators of the fraction in brackets, could we have it without, say, \\tilde{k}_x^4 in case of the second equation? In this case in the first equation the sign minus from the numerator could appear in front of the fraction?

5. Please reformulate the sentences start with a formula, e.g. “τ is an arbitrary time variable, consistent with ray approximations” (page 3, line 78), could become “Here τ…”.

Reviewer #2: In this article, the authors investigate the propagation of elastic waves along the axis of cylindrical shells with geometric imperfections and spatial variations in properties. They report the existence of branched flows of flexural waves in such waveguides and derive the scaling laws from the ray equations. They showed that the numerical integration of the ray equations and finite element numerical simulations are consistent with the theoretically derived scaling. They also note a universality for the exponents in the scaling with respect to similar observations in the past for waves in other physical contexts, as well as dispersive flexural waves in elastic plates. The authors suggest an immediate extension of this work on cylindrical elastic shells to explore the dependence of the scaling of the first caustic with the radius for shells with appreciable curvature. However, they are unable to do so in this work due to the limitation of 2πR ≳ Lc. They suggest using anisotropic randomness to enable reducing the radius by using a smaller correlation length in the circumferential direction and exploring an elegant scaling of 〈lf 〉 with radius in this parameter regime.

Overall, this article presents a well-structured and well-executed study on the propagation of elastic waves in cylindrical shells with random spatial variations. The authors' findings are significant and contribute to the current understanding of branched flows in wave propagation through random media. The article provides theoretical, numerical, and analytical evidence to support the authors' claims and suggests future directions for research, and may be accepted for the publication once it is comprehended to answers the following question.

1) What are the practical applications of understanding the branched flows of flexural waves in cylindrical shells, and how can this knowledge be utilized in real-world scenarios?

2) Can the findings of this study be generalized to other types of structures or materials, or is it specific to cylindrical shells with correlated random properties?

3) What are the limitations of the numerical integration of the ray equations and the full FE elastodynamic simulations, and how accurate are these methods in predicting the behavior of elastic waves in cylindrical shells?

4) What are the potential implications of using anisotropic randomness to reduce the radius of cylindrical shells, and how would this affect the scaling of 〈lf〉 with radius in this parameter regime?

5) How can the results of this study be applied to improve the design and engineering of cylindrical shell structures, and what further research is needed to fully understand the behavior of elastic waves in such structures?

Some minor changes are:

1) On page 2, line 88, Eq (3) should be Eq (2).

2) On page 4, line 94 (details in Methods) can be deleted, and likewise details provided in rest of the article with figures and sentences.

6. PLOS authors have the option to publish the peer review history of their article (what does this mean?). If published, this will include your full peer review and any attached files.

Reviewer #1: No

Reviewer #2: No

---

## [Author Response · Author response to Decision Letter 0]

22 Apr 2023

Responses to Reviewer Comments are in the Rebuttal Letter.

---

## [Editor Report · Decision Letter 1]

16 May 2023

Branched flows of flexural elastic waves in non-uniform cylindrical shells

PONE-D-23-04637R1

Dear Dr. Atul Bhaskar

We’re pleased to inform you that your manuscript has been judged scientifically suitable for publication and will be formally accepted for publication once it meets all outstanding technical requirements.

Kind regards,

Rab Nawaz

Academic Editor

PLOS ONE

Additional Editor Comments (optional):

After careful consideration of the revised manuscript and the responses provided by the authors, I am satisfied that the revised manuscript has successfully addressed the minor revisions suggested by the reviewers.
---

## [Editor Report · Acceptance letter]

18 May 2023

PONE-D-23-04637R1 

Branched flows of flexural elastic waves in non-uniform cylindrical shells  

Dear Dr. Bhaskar:

I'm pleased to inform you that your manuscript has been deemed suitable for publication in PLOS ONE. Congratulations! Your manuscript is now with our production department. 

Kind regards, 

on behalf of

Dr. Rab Nawaz 

Academic Editor

PLOS ONE